# Costs and Factors Associated with Hospitalizations Due to Severe Influenza in Catalonia (2017–2020)

**DOI:** 10.3390/ijerph192214793

**Published:** 2022-11-10

**Authors:** Mercè Soler-Font, Ignacio Aznar-Lou, Luca Basile, Núria Soldevila, Pere Godoy, Ana Martínez, Antoni Serrano-Blanco, Angela Domínguez

**Affiliations:** 1PRISMA Research Group, Institut de Recerca Sant Joan de Déu, Santa Rosa 39-57, 08950 Esplugues de Llobregat, Spain; 2Consortium for Biomedical Research in Epidemiology & Public Health (CIBERESP), Monforte de Lemos 3-5, 28029 Madrid, Spain; 3Public Health Agency of Catalonia, Department of Health, Roc Boronat, 81-95, 08005 Barcelona, Spain; 4Medicine Department, Universitat de Barcelona, Casanova 143, 08036 Barcelona, Spain; 5Institut de Recerca Biomèdica de Lleida, IRB Lleida, Alcalde Rovira Roure 80, 25198 Lleida, Spain; 6Parc Sanitari Sant Joan de Déu, Doctor Antoni Pujadas 42, 08830 Sant Boi de Llobregat, Spain

**Keywords:** influenza vaccination, comorbidity, hospitalization, hospital costs

## Abstract

This study aimed to estimate the cost and factors associated with severe hospitalized patients due to influenza in unvaccinated and vaccinated cases. The study had a cross-sectional design and included three influenza seasons in 16 sentinel hospitals in Catalonia, Spain. Data were collected from a surveillance system of influenza and other acute respiratory infections. Generalized linear models (GLM) were used to analyze mean costs stratified by comorbidities and pregnancy. Multivariate logistic models were used to analyze bacterial coinfection, multi-organ failure, acute respiratory distress syndrome, death and ICU admission by season and by vaccination status. Costs of ICU, hospitalization and total mean costs were analyzed using GLM, by season and by vaccination status. All models were adjusted for age and sex. A total of 2742 hospitalized cases were included in the analyses. Cases were mostly aged ≥ 60 years (70.17%), with recommended vaccination (86.14%) and unvaccinated (68.05%). The ICU admission level was statistically significant higher in unvaccinated compared to vaccinated cases. Costs of cases with more than or equal to two comorbidities (Diff = EUR − 1881.32), diabetes (Diff = EUR − 1953.21), chronic kidney disease (Diff = EUR − 2260.88), chronic cardiovascular disease (Diff = EUR − 1964.86), chronic liver disease (Diff = EUR − 3595.60), hospitalization (EUR 9419.42 vs. EUR 9055.45), and total mean costs (EUR 11,540.04 vs. 10,221.34) were statistically significant higher in unvaccinated compared to vaccinated patients. The influenza vaccine reduces the costs of hospitalization. There is a need to focus strategies in recommended vaccination groups.

## 1. Introduction

Influenza is an acute viral infection of the upper or lower respiratory tract, marked by fever, chills, and a generalized feeling of malaise. Most people who become ill with influenza virus infection recover without serious complications or sequelae. However, influenza can be associated with serious illness, hospitalization, and death, particularly among older adults, very young children, pregnant women, and persons of all ages with certain chronic medical conditions [1].

Influenza causes annual epidemics that affect between 5 and 15% of the world population. It is estimated that between 3 and 5 million people have serious influenza with complications that require hospitalization annually, and approximately 250,000–500,000 people die each year due to influenza [2]. Influenza may affect people of any age, but complications and hospitalizations are more frequent in childhood, especially in children aged < 5 years, people with chronic disease, and people aged ≥ 65 years [3,4]. Older age is a risk factor for intensive care unit (ICU) admission and death [5]. Seasonal influenza epidemics are responsible for a considerable clinical and economic burden [6]. Several studies in European high-income countries have reported influenza epidemics costing a mean of EUR 327/person [6], but hospitalized cases impose substantial costs ranging from EUR 6100 to EUR 8300 [7]. Vaccination is the most effective strategy to control influenza epidemics. Studies have shown that the number of deaths can decrease by 75% with vaccination, and the number of hospitalizations by up to 39% [8]. This is important not only for the patients, but also for resource allocation in epidemics since how to allocate limited resources is an ongoing problem of public policies, which highlights the importance of developing health cost studies [9]. Nevertheless, there is a lack of studies that estimate the costs of hospitalized influenza cases by comparing unvaccinated and vaccinated patients.

Therefore, this study aimed to estimate the cost of severe hospitalized cases due to influenza and associated factors in unvaccinated and vaccinated cases during three influenza seasons in the 16 sentinel hospitals of Catalonia.

## 2. Materials and Methods

This study was designed as a cross-sectional study during three influenza seasons (2017–2018, 2018–2019 and 2019–2020) in sentinel hospitals in Catalonia.

### 2.1. Study Context and Population

Catalonia is a Spanish region that has 7.5 million inhabitants. The health system is public with universal coverage (i.e., all inhabitants are guaranteed free health care).

A surveillance system for influenza and other acute respiratory infections (PIDIRAC) was started in the 1999–2000 season, based on a network of sentinel primary care physicians (general practitioners and pediatricians) who report influenza-like activity detected in their reference population daily [10]. Due to the situation generated by the 2009 pandemic caused by the new influenza A (H1N1) 2009 virus, the PIDIRAC sentinel network included the surveillance of severe hospitalized cases of laboratory-confirmed influenza in order to assess the severity of the pandemic and, from 2010–2011 onward, it was included as part of the surveillance system [11]. The hospital-based surveillance provides supplementary information to the sentinel surveillance from primary healthcare and allows for the assessment of the severity of the infection by studying case symptomatology, demographic characteristics, medical conditions, and vaccination status.

Catalonia has 16 sentinel hospitals that monitor 73.82% (n = 5,610,858) of the population. From this source of severe hospital influenza patients, we included laboratory-confirmed cases of influenza virus requiring hospitalization due to severity (pneumonia, acute respiratory distress syndrome, septic shock, multi-organ failure, or any other serious condition, including ICU admission and death), as well as those who developed this condition during hospitalization for another reason.

### 2.2. Variables

Information was recorded in a structured questionnaire used by physicians to collect data from each reported case by interview and the review of medical records.

The variables included were sex, age, comorbidities (asthma, obesity, chronic obstructive pulmonary disease, diabetes, chronic kidney disease, immunodeficiency, chronic cardiovascular disease, chronic liver disease and others ashemoglobinopathy, severe neuromuscular disease, or cognitive dysfunction), pregnancy, clinical complications (bacterial coinfection, multi-organ failure, acute respiratory distress syndrome), final status (death or recovery), clinical progress (date of hospital admission and discharge, ICU admission, number of days in ICU), and influenza vaccination status (vaccinated/not vaccinated during current season).

Age was arranged into three categories (<18 years, 18–59 years, ≥60 years) taking into account that the recommended vaccination for influenza is age 60 and over in Catalonia. A categorical variable with the number of comorbidities was created considering three categories: no comorbidities, one comorbidity, and two or more comorbidities. A recommended vaccination variable was created including all patients aged ≥ 60 years old, and/or those with one or more comorbidities.

### 2.3. Costs

Costs were considered from a hospital perspective. The hospital perspective included direct health costs of the hospital, to which we added the cost of vaccination. All costs were expressed in euros (EUR) for the reference year 2020.

Unit costs were obtained from the official rates published by the public provider (Catalan Service of Health) [12] (Table 1). Hospitalization costs were calculated by multiplying the number of days hospitalized (differentiating between days 1 to 5 and subsequent days) by their unit cost and totalling the cost of the hospital center according to the level of complexity. The levels of complexity were classified into the 6 categories established by the Catalan Government, ranging from level 1 (i.e., complementary hospital) to level 6 (i.e., high complexity hospital). In this study, the sentinel hospitals range from level 4 to 6 [12]. ICU stay costs were calculated by multiplying the number of days in ICU by their unit cost. Costs of influenza vaccination were calculated as the sum of the cost of the vaccine and the unit cost of the nursing visit for vaccination. The cost of the vaccine was obtained from the *Barcelona College of Pharmacists (COFB)* internal database [13].

### 2.4. Statistical Analysis

Descriptive analyses by season were performed for sex, age, recommended vaccination, and vaccination status. Additionally, descriptive analyses between vaccinated and unvaccinated cases for the whole period were also carried out for sex, age, and recommended vaccination.

Generalized linear models (GLM) with family gamma and log link, adjusted for sex and age, were carried out to compare the total mean costs of cases with comorbidities or pregnancy in the vaccinated and the unvaccinated cases.

Multivariate logistic regression models were made to analyze the differences in clinical complications (bacterial coinfection, multi-organ failure, and acute respiratory distress syndrome, ARDS), final status (death) and ICU admission by season, taking 2017–2018 as reference, and between vaccinated and unvaccinated cases for the whole period adjusted for sex, age, and recommended vaccination. GLM with family gamma and log link were carried out to examine the differences in clinical progress (hospitalization costs and mean costs of ICU stay) and total mean costs by season and by vaccination status, adjusted for sex, age, and recommended vaccination.

Two hundred and five cases had missing data for hospitalization costs and total mean costs, and were not included in the costs analyses.

The analyses were performed using the STATA 13 statistical package.

## 3. Results

The characteristics of the 2742 hospitalized cases included in the analyses are shown in Table 2. Of the hospitalized cases, 1299 pertained to 2017–2018, 1106 to 2018–2019, and 337 to 2019–2020. From season 2017–2018 to 2019–2020, cases under 18 years old increased (4.85% in 2017–2018, and 9.79% in 2019–2020), and those over 60 years old decreased (74.44% in 2017–2018, and 63.80% in 2019–2020). In addition, there was a decrease in recommended vaccination and vaccination status between 2017–2018 and 2019–2020.

For vaccination status, Table 3 shows that 2362 hospitalized cases had recommended vaccination, and of them, only 863 were vaccinated (36.54%). Of those who had no recommendation for vaccination (n = 380), 13 were finally vaccinated (3.48%).

The comorbidities and mean costs of hospitalized influenza cases by vaccination status are shown in Table 4. In total, 37.73% of the unvaccinated had two or more comorbidities, compared to 61.07% of the vaccinated, but the mean cost was statistically significant higher in the unvaccinated compared to vaccinated (Diff = EUR − 1881.32). For diabetes (EUR − 1953.21), chronic kidney disease (Diff = EUR − 2260.88), chronic cardiovascular disease (Diff = EUR − 1964.86), chronic liver disease (Diff = EUR − 3595.60) and others as hemoglobinopathy, severe neuromuscular disease or cognitive dysfunction (Diff = EUR − 4340.85), the costs were statistically significant lower in the vaccinated group than in the unvaccinated group. No statistically significant differences were observed between the vaccinated and the unvaccinated in terms of asthma, obesity, COPD, immunodeficiency, and pregnancy.

The clinical complications and economic costs for unvaccinated and vaccinated cases by season, and overall, are shown in Table 5.

Bacterial coinfection showed a statistically significant decrease in the 2018–2019 and 2019–2020 seasons (31.19% and 28.07%, respectively) compared with the 2017–2018 season (47.98%). ARDS showed a statistically significant increase in 2018–2019 (52.50%) and a decrease in 2019–2020 (38.84%) compared with the 2017–2018 season (45.57%). Multi-organ failure, final status (death), clinical progress (ICU admission, mean cost per ICU stay, and cost of hospitalization), and total monetary cost did not show statistically significant differences in vaccinated cases by season.

By vaccination status, 21.54% of unvaccinated cases were admitted to the ICU compared to 13.13% of those vaccinated (*p* < 0.001). Hospitalization costs were statistically significant higher in unvaccinated (EUR 9419.42) than vaccinated cases (EUR 9055.45), and total mean costs were statistically significant higher in unvaccinated (EUR 11,540.04) than vaccinated (EUR 10,221.34) cases. Nevertheless, clinical complications and final status (death rate) did not show statistically significant differences.

## 4. Discussion

This study shows that total mean hospitalization costs were statistically significant higher for unvaccinated patients than those who were vaccinated. ICU admission was statistically significant higher in the unvaccinated than in the vaccinated, although there were no statistically significant differences when comparing the mean costs of those that were admitted to ICU. Costs of patients with two or more comorbidities, diabetes, chronic kidney disease, chronic cardiovascular disease, chronic liver disease, and others (i.e., hemoglobinopathy, severe neuromuscular disease or cognitive dysfunction) were statistically significant higher in unvaccinated than in vaccinated patients. By season, bacterial coinfection showed a statistically significant decrease in vaccinated cases in 2018–2019 and 2019–2020 compared with 2017–2018. The sociodemographic characteristics showed statistically significant decreases in age, recommended vaccination, and vaccination status by season. By vaccination status, vaccinated cases were older than those unvaccinated. Overall, 63.3% of the cases should have been vaccinated but were not.

### 4.1. Comparison with the Existing Literature

Total mean costs were statistically significant higher for unvaccinated patients (EUR 12,002.56) than those vaccinated (EUR 10,650.66). This result coincides with Storch et al. (2021), who found cost-saving effects for influenza vaccination in hospitalized cases [14]. In addition, ICU admission was statistically significant higher in the unvaccinated than in vaccinated patients. This result agrees with other authors who observed that the influenza vaccination had a protective effect in terms of disease severity [15,16]. Moreover, only 36.54% of cases with recommended vaccination were vaccinated (i.e., most of the cases that had a higher risk of complications were not vaccinated). Vaccine hesitancy could be a reason for this low adherence to vaccination, especially in those cases with recommended vaccination [17].

Costs of patients with diabetes, chronic kidney disease, chronic cardiovascular disease, chronic liver disease, and those with two or more comorbidities were statistically significant higher in unvaccinated than in vaccinated patients. These comorbidities are included in the recommended vaccination criteria in Catalonia, as they can lead to a high risk of complications from influenza [18,19] and, consequently, a higher cost. Coinciding with our study, the systematic review by Federici et al. (2018) found that co-morbidities and complications, especially diabetes and chronic cardiovascular disease, had an important impact on costs [20].

Bacterial coinfection showed a statistically significant decrease in the 2018–2019 and 2019–2020 seasons compared with the 2017–2018 season. This decline might be due, at last in part, to the implementation of strategies to combat antibiotic resistance. Specifically, in Spain, from June 2017, the “Strategic and Action Plan to Reduce the Risk of Selection and Dissemination of Antimicrobial Resistance 2014–2018” was implemented, which included several actions and recommendations [21,22,23], and between 2018 and 2020 more recommendations were developed and implemented [24,25,26,27]. In 2018–2019, ARDS showed a statistically significant increase compared to 2017–2018, possibly explained by differences in the type of influenza virus in hospitalized cases. In 2018–2019, 99% of hospitalized cases were influenza A virus, which could involve greater risk of developing ARDS, while in 2017–2018 it was 40% [28,29]. ARDS showed also a decrease in the year 2019–2020 compared to 2017–2018, possibly related to the percentage of vaccination in hospitalized cases due to influenza A virus being higher than in 2017–2018 [29].

By seasons, age, percentage of cases with recommended vaccination, and those vaccinated showed a statistically significant decrease. The increasing trend of influenza cases in Catalonia in the population under 18 years of age may explain the increase in hospitalized cases in this age range and, therefore, the decrease in the distribution of age, as well as the decrease in recommended vaccination and those vaccinated [29,30].

Finally, 63.46% of the cases with recommended vaccination were not vaccinated. These findings should be considered, especially when the total costs and complications of the unvaccinated are taken into account, in the effort to improve vaccination strategies and achieve higher vaccination coverage in the population at risk, and to get closer to the goal set by the WHO of achieving coverage of at least 75% in elderly people [15,31].

### 4.2. Implications for Practice and Research

This study calculated the costs of hospitalized cases of influenza and associated factors, comparing unvaccinated and vaccinated patients. We found that hospitalizations for influenza in Catalonia had a higher cost for the unvaccinated, and this group also accounted for more ICU admissions. In addition, 63.3% of cases admitted for influenza had been recommended vaccination (i.e., ≥60 years and/or comorbidities) but did not get vaccinated. Therefore, policy makers need to focus their efforts on developing strategies to increase vaccination rates in the groups for whom vaccination is recommended.

Future research should examine other factors that may influence patients who appear to be healthy (i.e., without recommended vaccination) yet end up in hospital. Studies should evaluate not only the hospitalization costs, but also primary care costs and productivity losses for patients and caregivers.

### 4.3. Limitations

This study has several limitations. First, we did not have data on primary care or sick leave, which could limit the perspective of analysis and cost estimation. Second, the variable mean costs per ICU stay had approximately 40% of the data missing, so for the cost calculation we excluded the cases with this large volume of missing data. Therefore, costs may have been underestimated. Third, only 13% of cases were from the 2019–2020 season. This season saw the epidemic start in the third week of January, and it lasted eight weeks, coinciding with the beginning of the COVID-19 pandemic [29]. Thus, it is possible that SARS-CoV-2 could have displaced the influenza epidemic or that the surveillance system was already focused on COVID-19, with a concomitant underreporting of influenza cases by physicians. Finally, there were few data on cases under 18 years old, so results for this age group should be approached with caution.

## 5. Conclusions

Unvaccinated patients had higher total mean hospitalization costs and ICU admission than vaccinated cases. Unvaccinated patients with comorbidities also had higher costs than those who were vaccinated. Bacterial coinfection showed a decrease by season. In total, 63.46% of cases admitted to hospital for influenza in Catalonia had been recommended vaccination but did not get vaccinated. Policy makers need to consider developing strategies to increase vaccination rates in groups for whom vaccination is recommended.

## Figures and Tables

**Table 1 ijerph-19-14793-t001:** Unit costs of hospital discharge, ICU stay, hospitalization and influenza vaccination.

Tariffs	Unit Cost
Hospital centre (EUR/discharge) ^1^	
Level 4: Reference hospital type A	EUR 1299.91
Level 5: Reference hospital type B	EUR 1772.08
Level 6: High complexity hospital	EUR 2233.19
ICU stay (EUR/day)	EUR 1550.00
Hospitalization (EUR/day)	
Day 1–5	EUR 751.00
Day > 5	EUR 597.00
Influenza vaccination	EUR 56.60

^1^ Sentinel hospitals in this study range from level 4 to 6.

**Table 2 ijerph-19-14793-t002:** Demographic characteristics of cases by season and vaccination status.

	Total (n = 2742)		Season						Vaccination Status			
			2017–2018 (n = 1299)		2018–2019 (n = 1106)		2019–2020 (n = 337)		Unvaccinated (n = 1866)		Vaccinated (n = 876)	
**Sex**	**n**	**%**	**n**	**%**	**n**	**%**	**n**	**%**	**n**	**%**	**n**	**%**
Female	1229	44.82	560	43.11	521	47.11	148	43.92	839	44.96	390	44.52
Male	1513	55.18	739	56.89	585	52.89	189	56.08	1027	55.04	486	55.48
**Age**												
<18 years	191	6.97	63	4.85	95	8.59	33	9.79	183	9.81	8	0.91
18–59 years	627	22.87	269	20.71	269	24.32	89	26.41	570	33.55	57	6.51
≥60 years	1924	70.17	967	74.44	742	67.09	215	63.80	1113	59.65	811	92.58
**Recommended vaccination**												
Yes	2362	86.14	1162	89.45	927	83.82	273	81.01	1499	80.33	863	98.52
No	380	13.86	137	10.55	179	16.18	64	18.99	367	19.67	13	1.48
**Vaccination**												
Yes	876	31.95	445	34.26	340	30.74	91	27.00	-	-	-	-
No	1866	68.05	854	65.74	766	69.26	246	73.00				

**Table 3 ijerph-19-14793-t003:** Recommended vaccination by vaccination status.

	Total (n = 2742)	Unvaccinated (n = 1866)		Vaccinated (n = 876)	
	n	n	% ^1^	n	% ^1^
**Recommended**	2362	1499	63.46	863	36.54
**Not recommended**	380	367	96.58	13	3.48

^1^ Percentages calculated using as denominator n = 2362 for recommended vaccination cases, and n = 380 for not recommended vaccination cases.

**Table 4 ijerph-19-14793-t004:** Medical conditions and mean costs of hospitalized influenza cases by vaccination status.

	Unvaccinated (n = 1866)				Vaccinated (n = 876)						Cost Difference	
			Costs (EUR)					Costs (EUR)			Costs (EUR)	
**Comorbidities (nº)**	**n**	**%**	**M**	**SD**	**MD**	**N**	**%**	**M**	**SD**	**MD**	**Diff ^2^**	**CI95%**
No	571	30.60	9582.22	9533.84	55	98	11.19	9516.21	9990.68	4	−1068.13	−3326.18 to 1189.93
1	591	31.67	11,669.88	12,449.26	39	243	27.74	10,098.08	10,434.89	11	−1718.55	−3453.74 to 16.64
≥2	704	37.73	12,992.92	13,990.13	58	535	61.07	10,412.25	11,806.73	38	**−1881.32**	−3194.98 to −567.66
**Comorbidities (%)**												
Asthma	150	8.17	9603.53	12,707.85	12	64	7.45	8677.23	5839.71	64	−1638.77	−5477.39 to 2199.86
Obesity	134	7.24	11,666.41	11,023.97	17	68	7.79	11,984.66	14,738.41	5	276.88	−3146.40 to 3700.15
COPD	374	20.14	12,682.69	12,482.13	27	289	33.03	11,231.8	13,993.17	21	−1341.46	−3141.94 to 459,02
Diabetes	395	21.33	12,491.83	12,659.99	41	279	32.03	10,111.83	7963.40	23	**−1953.21**	−3485.52 to −420.90
CKD	249	13.37	12,370.59	11,875.64	15	212	24.23	9772.02	6968.74	17	**−2260.88**	−3900.18 to −621.58
Immunodeficiency	303	16.33	13,789.71	13,343.33	19	150	17.24	11,978.67	19,176.18	14	−1157.92	−3707.74 to 1391.89
CCD	519	27.89	13,044.11	14,667.94	42	419	48.00	10,453.83	10,073.07	25	**−1964.86**	−3330.11 to −599.62
CLD	89	4.80	16,587.9	15,407.42	5	47	5.40	11,669.29	8603.26	4	**−3595.60**	−7151.05 to −40.16
Others ^1^	218	11.76	14,053.96	17,225.42	17	138	15.79	9238.17	11,129.94	10	**−4340.85**	−6892.83 to −1788.86
**Pregnancy**	10	0.54	13,789.93	23,633.5	1	3	0.34	5063.24	736.02	1	−10,89.16	−34,930.40 to 13,147.24

M: mean; SD: standard deviation; MD: missing data for costs; COPD: chronic obstructive pulmonary disease; CKD: chronic kidney disease, CCD: chronic cardiovascular disease; CLD: chronic liver disease; ^1^ Others: hemoglobinopathy, severe neuromuscular disease or cognitive dysfunction. ^2^ Diff: difference of costs between vaccinated and non-vaccinated calculated with a generalized linear model using family gamma and log link, adjusted for sex and age. In bold: *p*-value < 0.05.

**Table 5 ijerph-19-14793-t005:** Clinical complications and economic costs in vaccinated and unvaccinated cases by season and overall.

	Season								Vaccination Status (2017–2020)				
	2017–2018 (n = 1299)		2018–2019 (n = 1106)		P ^3^	2019–2020 (n = 337)		P ^3^	Unvaccinated (n = 1866)		Vaccinated (n = 876)		P ^4^
**Clinical Complications**	**n**	**%**	**n**	**%**		**n**	**%**		**n**	**%**	**n**	**%**	
Bacterial coinfection	380	47.98	175	31.19	<0.001	48	28.07	<0.001	418	38.99	185	40.93	0.933
Multi-organ failure	92	7.08	90	8.24	0.168	21	6.48	0.954	129	7.00	74	8.49	0.783
ARDS	592	45.57	578	52.50	<0.001	127	38.84	0.067	868	46.82	429	49.14	0.906
**Final status (death)**	171	13.16	138	12.48	0.811	43	12.76	0.645	210	11.25	142	16.21	0.252
**Clinical progress**													
ICU admission	216	16.63	227	20.52	0.031	74	21.96	0.062	402	21.54	115	13.13	<0.001
ICU stay mean cost ^1^ (m; SD)	14,378.18	15,938.22	14911	13,513.99	0.052	16,820.37	26,299.82	0.069	14,539.00	16,633.47	15,740.52	15,086.04	0.740
Hospitalization ^2^ (m; SD)	9476.55	9050.27	9331.37	7956.03	0.988	8434.92	12,372.63	0.123	9419.42	8883.67	9055.45	9544.39	0.017
**Total mean costs ^2^ (m; SD)**	11,551.69	12,415.77	10,869.18	10,120.45	0.296	10,016.18	15,605.95	0.058	11,540.04	12,362.48	10,221.34	11,228.97	<0.001

ARDS: acute respiratory distress syndrome. ^1^ Only patients admitted to ICU were considered. ^2^ The 205 missing data (MD) for hospitalization and total mean costs were not included in the analysis and were distributed as follows: season 2017–2018 (MD = 33), season 2018–2019 (MD = 125), season 2019–2020 (MD = 47), unvaccinated (MD = 152), and vaccinated (MD = 53). ^3^ Comparison with the reference season 2017–2018.^4^ Comparison between unvaccinated and vaccinated patients. For clinical complications, final status and ICU admission multivariate logistic models were used. For cost variables (ICU stay mean cost, hospitalization and total mean costs), generalized linear models using family gamma and log link were used. All were adjusted for sex, age, and recommended vaccination.

## Data Availability

The datasets used and/or analyzed during the current study are available from the corresponding author on reasonable request.

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
