# Peer review of "Costs and Factors Associated with Hospitalizations Due to Severe Influenza in Catalonia (2017–2020)"

_ijerph, 2022, doi:10.3390/ijerph192214793_

Round 1
Reviewer 1 Report
GENERAL CONSIDERATIONS
Though the costs of influenza epidemics are far from being a new topic, this study is interesting and contains useful information for many readers.
The design of the study is appropriate and well described, the results are clearly presented and discussed, and the limitations thoroughly acknolewdged.
Only a few amendments are needed (see below). They are especially necessary for some tables. Indeed I suggest to check all of them with attention.
SPECIFIC CONSIDERATIONS
I am not sure I found the ethics committee appproval
Abstract, lines 28-31 and Introduction lines 50-54: These two phrases are a bit difficult to understand and should be rephrased more clearly.
Table 1: Level 1, Level 2 and Level 3 are lacking
Table 2a: In the "Vaccination status" column I do not understand the percentages of the last four lines. For example in first line (Age <18 years) there are 191 subjects.. Of these 183 are unvaccinated and 8 vaccinated: how is it possible that their percentages are 9.81 and 0.91 respectively, considering that 191 is 100%? The same happens in the 3 lines below.
Table 2b. I understand that total patients who had not a recommendation for vaccination are 380, but at line 167 you said that they were 355. Of these you report in the table that 13 were vaccinated (3.48%), but at line 167 you said that 12 patients (3.4%) were vaccinated. Which is correct?
Among those in whom vaccination was recommended the table shows that eventually 863 were vaccinated. I understand they are the same 863 of table 2 a. If it is so why the percentage is 89.52% in table 2a and 35.64% in table 2b?
Finally, is table 2b really necessary?
Discussion.4.1. In lines 262 to 266 you introduce the problem of low adherence to vaccination. This is very important. I would also add a phrase to introduce one of the main causes of this phenomenon, i.e. vaccine hesitancy.
Reference 16: Is the spelling of Col.legi correct?
Is there a reference N. 34?
Reviewer 2 Report
See attachment

Round 2
Reviewer 2 Report
Authors addressed comments submitted.